# Association between Serum Selenium Levels and Lipids among People with and without Diabetes

**DOI:** 10.3390/nu15143190

**Published:** 2023-07-19

**Authors:** Qi Nie, Changsi Wang, Li Zhou

**Affiliations:** Department of Nutrition, Hygiene and Toxicology, Academy of Nutrition and Health, School of Public Health, Medical College, Wuhan University of Science and Technology, Wuhan 430065, China; nieqi@wust.edu.cn (Q.N.); wangchangsi@wust.edu.cn (C.W.)

**Keywords:** selenium, diabetic status, lipids, interaction, NHANES

## Abstract

The current study aimed to examine the association between serum selenium levels and lipids and explore whether the association was modified by diabetic status. A total of 4132 adults from the National Health and Nutrition Examination Survey (2011–2016) were included in this study. Multiple linear regression models were used to estimate the association between serum selenium and lipids. Higher serum selenium levels were significantly associated with increased total cholesterol (TC) (*p* < 0.001), triglyceride (TG) (*p* = 0.003), and low-density lipoprotein cholesterol (LDL-C) (*p* = 0.003) in the overall population. Diabetic status interacted with serum selenium for TC and LDL-C (*p* for interaction = 0.007 and <0.001). Comparing the highest with the lowest tertiles of serum selenium, the multivariate-adjusted β coefficients (95% CIs) were 17.88 (10.89, 24.87) for TC, 13.43 (7.68, 19.18) for LDL-C among subjects without diabetes, but nonsignificant among those with diabetes. In US adults, the serum selenium was positively associated with lipids and the association was modified by diabetic status. Higher serum selenium levels were significantly associated with increased TC and LDL-C among participants without diabetes, but not among participants with diabetes.

## 1. Introduction

Dyslipidemia is one of the main risk factors for cardiovascular disease, and the management of serum lipids has become a central goal for the prevention of cardiovascular disease [1]. To achieve this goal, identifying factors affecting lipid levels is of great importance. Emerging evidence suggests that nutrients are crucial for lipid metabolism [2,3].

Selenium, as an essential trace element, plays an important role in lipid metabolism [4]. Findings from in vitro and animal studies have demonstrated that high selenium status could induce mitochondrial dysfunction [5,6,7], dysregulate the expression of microRNA related to lipid metabolism [8,9], and upregulate the expression of key lipogenesis genes while suppressing the expression of lipolysis genes, which eventually lead to increased lipid levels [10]. However, findings from epidemiological studies examining the association between selenium and lipid levels are inconsistent. For instance, a study conducted in a rural Chinese population showed that elevated selenium was associated with elevated total cholesterol (TC) and low-density lipoprotein cholesterol (LDL-C) [11]. However, another study conducted in the Netherlands found no significant association between selenium status and TC and LDL-C [12].

Furthermore, previous studies have found that diabetic patients are more likely to have a lipid metabolism disorder [13]. Meanwhile, considering that glucose metabolism disorders significantly affect the metabolism of trace elements, including selenium [14], it could be speculated that diabetic status might modify the association between selenium level and lipids. However, to our knowledge, no study has evaluated the potential modifications by diabetic status on the association between selenium and lipids.

Therefore, based on a nationally representative sample of participants from the National Health and Nutrition Examination Survey (NHANES) (2011–2016), we conducted this study to explore the association between serum selenium and lipids in the overall population as well as participants with and without diabetes and to assess whether the selenium-lipid association was modified by diabetic status. 

## 2. Methods

### 2.1. Study Design and Participants

The NHANES is a population-based cross-sectional survey designed to gather information about the health and nutrition of the US population. The survey, conducted by the National Center for Health Statistics, combines interviews and physical examinations and provides comprehensive demographic, socioeconomic, dietary, medical, dental, physiological, laboratory, and health-related information. More details on subject recruitment, survey design, and data collection are available online. The NHANES survey was approved by the National Center for Health Statistics Research Ethics Review Board, and written informed consent was obtained from all participants.

In the 2011–2016 NHANES study, we included participants who were aged above 20 years old and completed the measurements of serum selenium levels (*n* = 5153). After excluding those who were pregnant (*n* = 55), or whose diabetic status could not be defined (*n* = 966), a total of 4132 subjects were included in the current study.

### 2.2. Definition of Diabetes

According to the American Diabetes Association criteria [15], diabetes is defined by one of the following criteria: (1) glycosylated hemoglobin level ≥ 6.5%; (2) fasting plasma glucose level ≥ 126 mg/dL (7.0 mmol/L); (3) 2 h post-challenged plasma glucose level ≥ 200 mg/dL (11.1 mmol/L).

### 2.3. Measurement of Serum Selenium Levels

Blood specimens were collected in vacuum containers screened for selenium contamination. After coagulation and centrifugation, serum samples were collected and stored under appropriate frozen conditions (−20 °C) until shipped to the National Center for Environmental Health for analysis. Serum selenium was measured by inductively coupled plasma dynamic reaction cell mass spectrometry. The between-assay coefficients of variation between pools were 2.5 to 4.4%. The lower limit of detection (LLOD) was 4.5 µg/L. Serum selenium levels below the LLOD were imputed as the LLOD divided by the square root of 2.

### 2.4. Measurement of Serum Lipids Levels

Fasting morning venous blood specimens were collected for the determination of triglyceride (TG) and LDL-C. Serum specimens were stored under appropriate frozen conditions (−80 °C) until shipped to the University of Minnesota for analysis. TC and TG were measured by enzymes, high-density lipoprotein cholesterol (HDL-C) was measured by immunoassay, and LDL-C was calculated by the Friedewald equation: LDL-C = TC-HDL-C-TG/5. 

### 2.5. Assessment of Other Covariates

Information on age, sex, race/ethnicity, education, family income to poverty ratio, marital status, smoking status, drinking status, and physical activity (PA) were self-reported during standard questionnaire interviews. Race/ethnicity was categorized as Mexican American, non-Hispanic White, non-Hispanic Black, or others. Education was categorized as less than high school, high school graduate/GED or equivalent, or some college or above. Family income to poverty ratio was categorized as 0–1.0, 1.0–3.0, or >3.0. Marital status was categorized as single, married (married or living with a partner), or others (widowed, divorced, or separated). Smoking status was categorized as non-smoker (had < 100 cigarettes during their lifetime), former smoker (had ≥ 100 cigarettes during their lifetime and had quit smoking before the time of the study), or current smoker (had ≥ 100 cigarettes during their lifetime and was smoking at the time of the study). Drinking status was categorized as non-drinker (had < 12 drinks during their lifetime), former drinker (had ≥ 12 drinks during their lifetime or any one year, but none in the past 12 months), or current drinker (had ≥ 12 drinks during their lifetime, and drank ≥ 1 time in the past 12 months). PA was calculated by summing minutes of activity per week multiplied by the metabolic equivalent (MET) score of each activity. Based on the US PA guidelines, PA was categorized as inactive group (PA = 0 MET·min/week), moderate group (PA < 60 MET·min/week), or vigorous group (PA ≥ 60 MET·min/week) [16].

Anthropometric measurements and dietary assessments were performed by highly trained medical personnel. Body mass index (BMI) was calculated as weight (in kilograms) divided by the square of height (in meters). Healthy eating index-2015 (HEI-2015) was calculated to assess objective diet quality [17].

### 2.6. Statistical Analysis

Sample weights, clustering, and stratification were incorporated in all analyses to account for the complex sampling design of the NHANES. Characteristics of the population were presented as means (standard errors) for continuous variables or numbers (percentages) for categorical variables. Differences in characteristics between subgroups were estimated using the Chi-square test for categorical variables or one-way ANOVA for continuous variables. The serum selenium and lipids levels among participants overall and participants with and without diabetes are expressed as mean ± standard errors and illustrated through histograms. Comparison between different diabetic status groups was performed using the one-way ANOVA. The association between serum selenium and lipids level was estimated using multiple linear regression models, with adjustment for age, sex, race/ethnicity, education, family income to poverty ratio, marital status, BMI, smoking status, drinking status, PA, and HEI-2015. Serum selenium was treated as a continuous variable (per standard deviation (SD) increase) or a categorical variable (using the lowest tertile as the reference). To examine whether the associations between serum selenium and lipids differ by diabetic status, interaction tests with multiplicative terms between serum selenium and diabetic status were performed. All analyses were performed using the SAS software (version 9.4), and *p* values < 0.05 (two-sided) were considered statistically significant.

## 3. Results

### 3.1. Characteristics of the Study Participants

Characteristics of the overall study population, as well as stratified by diabetic status are shown in Table 1. The current study (weighted mean age 47.0 years, 51.0% male) included 782 individuals with diabetes and 3350 individuals without diabetes. Participants with diabetes were older, more likely to be a non-drinker, less likely to be a non-smoker, and performed less PA and education when compared with participants without diabetes.

The serum selenium and lipids levels among participants overall, as well as participants with and without diabetes, are shown in Figure 1. The mean (standard error) serum selenium level in the overall study population was 130.6 (0.6) µg/L. Participants with diabetes showed a higher level of selenium and TG, as well as a lower level of HDL-C compared to participants without diabetes. Moreover, there were no significant differences in TC and LDL-C between participants with diabetes and those without diabetes.

### 3.2. Association of Serum Selenium with TC

The results of the multiple linear regression showed that higher serum selenium levels were significantly associated with higher TC among participants overall, and participants without diabetes, but not in those with diabetes (Table 2). Among participants overall, the multivariate-adjustment β coefficient (95% CI) was 12.75 (6.02, 19.47). Among participants without diabetes, the multivariate-adjustment β coefficient (95% CI) was 17.88 (10.89, 24.87), when comparing the highest to the lowest tertiles of serum selenium levels. When serum selenium was treated as a continuous variable, similar results were observed. Among participants overall, the multivariate-adjustment β coefficient (95% CI) for TC per SD increase of serum selenium was 5.54 (3.34, 7.74). Among participants without diabetes, the multivariate-adjustment β coefficient (95% CI) for TC per SD increase of serum selenium was 6.89 (4.51, 9.26).

### 3.3. Association of Serum Selenium with TG

The results of the multiple linear regression showed that higher serum selenium levels were significantly associated with higher TG among participants overall, and participants without diabetes, but not in those with diabetes (Table 3). Among participants overall, the multivariate-adjustment β coefficient (95% CI) was 23.31 (8.22, 38.40). Among participants without diabetes, the multivariate-adjustment β coefficient (95% CI) was 22.39 (6.92, 37.86), when comparing the highest to the lowest tertiles of serum selenium levels. When serum selenium was treated as a continuous variable, similar results were observed. Among participants overall, the multivariate-adjustment β coefficient (95% CI) for TG per SD increase of serum selenium was 8.85 (2.68, 15.01). Among participants without diabetes, the multivariate-adjustment β coefficient (95% CI) for TG per SD increase of serum selenium was 5.57 (0.44, 10.70).

### 3.4. Association of Serum Selenium with HDL-C

The association between serum selenium and HDL-C in the overall study population, stratified by diabetic status, are shown in Table 4. Comparing the highest to the lowest tertiles of serum selenium levels, no significant association between serum selenium and HDL-C levels was observed in any subgroup stratified by diabetic status. When serum selenium was treated as a continuous variable, SD increase of serum selenium was not associated with HDL-C among participants overall, participants with or without diabetes. 

### 3.5. Association of Serum Selenium with LDL-C

The results of the multiple linear regression showed that higher serum selenium levels were significantly associated with higher LDL-C among participants overall, participants without diabetes, but not in those with diabetes (Table 5). Among participants overall, the multivariate-adjustment β coefficient (95% CI) was 8.87 (3.29, 14.46). Among participants without diabetes, the multivariate-adjustment β coefficient (95% CI) was 13.43 (7.68, 19.18), when comparing the highest to the lowest tertiles of serum selenium levels. When serum selenium was treated as a continuous variable, similar results were observed. Among participants overall, the multivariate-adjustment β coefficient (95% CI) for LDL-C per SD increase of serum selenium was 3.80 (1.52, 6.09). Among participants without diabetes, the multivariate-adjustment β coefficient (95% CI) for LDL-C per SD increase of serum selenium was 5.51 (3.02, 8.00).

### 3.6. Interaction between Serum Selenium and Diabetic Status for Lipids

Interactions analysis with multiplicative terms between serum selenium and diabetic status was performed to examine whether the associations between serum selenium and lipids differed by diabetic status (Table 6). After adjustment for age, sex, race/ethnicity, education, family income to poverty ratio, marital status, BMI, smoking status, drinking status, PA, and HEI-2015, diabetic status interacted with serum selenium for TC and LDL-C significantly (*p* for interaction = 0.007 and <0.001). However, no significant interaction was observed between diabetic status and serum selenium for TG or HDL-C (all *p* for interactions > 0.05).

## 4. Discussion

In this nationally representative sample of US adults, we observed that higher serum selenium levels were significantly associated with increased TC, TG, and LDL-C in the overall population. In addition, we also found that diabetic status interacted with serum selenium for TC and LDL-C. The serum selenium level was positively associated with TC and LDL-C among participants without diabetes. However, the positive association was not observed among participants with diabetes. The current study extends previous findings on the association between serum selenium and lipids. 

Similar to our findings, several cross-sectional studies also revealed that high selenium was associated with increased lipid levels in the general population [18,19]. For instance, one study conducted in a UK adult population found that higher plasma selenium was associated with increased TC [20]. The study by Huang et al. found that subjects with the highest quartile of circulating selenium had higher odds of elevated TC, TG, and LDL-C [21]. However, in another study of 82 healthy Dutch subjects, no significant association was observed between selenium and TC, LDL-C [12]. It is worth noting that the study is limited by a small sample size and should be interpreted with caution. So far, evidence from cohort studies on the association between selenium and lipids is limited. In a cohort study of 140 older Chinese adults over 65 years of age, Chen et al. found that individuals in the highest nail selenium quartile group showed a 1.11 SD decrease on TC and a 0.52 SD decrease on TG after seven years [22]. Considering that nails are on the body surface and are more likely to be contaminated by external selenium-containing substances, the mentioned findings should be interpreted with caution [23]. 

In the current study, we found diabetic status interacted with serum selenium for TC and LDL-C. Although such an interaction has not been reported before, several studies corroborate our findings. A randomized controlled trial conducted among a population with type 2 diabetes found that selenium supplementation did not change TC and LDL-C levels [24]. In addition, two studies conducted among populations with abnormal glucose metabolism also showed that selenium supplementation had no significant effect on lipid levels [25,26]. In the study conducted by Asemi et al., patients with gestational diabetes were given selenium (200 mg/day) or a placebo for six weeks, but no significant changes in lipid levels were observed in the intervention group [25]. Similar findings were observed among the population with diabetic nephropathy [26].

Several mechanisms may explain the association between selenium and lipids. Firstly, selenium may be involved in abnormal lipid metabolism by disturbing mitochondrial function. Zhuang et al. found that high selenium exposure induced mitochondrial dysfunction by increasing the generation of reactive oxygen species [5]. Mitochondrial dysfunction can cause endoplasmic reticulum stress and disrupt lipid metabolism by impairing autophagy [6,7]. Secondly, cholesterol 7 α-hydroxylase acts as a rate-limiting enzyme for bile acid biosynthesis, catalyzing the conversion of cholesterol into bile acid for elimination from the body. Guo et al. found that blood selenium was positively associated with the expression of microRNA-122 [8]. The expression of microRNA-122 reduced cholesterol 7 α-hydroxylase protein and increased cholesterol levels [9]. Finally, previous studies have found the interdependence of selenoprotein and lipoprotein metabolic pathways. Selenoprotein P is taken up by the brain and testes via the apolipoprotein E receptor-2, while another apolipoprotein receptor, megalin, mediates selenoprotein P uptake in the kidney [27,28]. In a mouse model of gene knockout associated with selenium protein synthesis, the expression of key genes involved in cholesterol biosynthesis, metabolism, and transport were also altered, which suggested a role for selenoproteins in the regulation of lipoprotein biosynthesis [29]. Nonetheless, more mechanism studies are warranted to further illustrate the potential mechanisms of the selenium-lipid relation.

So far, the mechanism underlying the interaction between selenium and diabetic status for lipids remains to be clarified. Findings from previous studies suggest that the metabolism of essential trace metals, such as selenium, are often impaired in populations with diabetes [14,30]. Compared with non-diabetic subjects, diabetic patients had significantly lower serum selenium levels, suggesting that diabetic patients have higher metabolic needs for selenium [31,32]. In this study, the difference in the selenium–lipid association between participants with and without diabetes might be explained by the increased demands of trace elements in people with diabetes.

Our study has several limitations. First, due to the cross-sectional design, the causal relationship between selenium and serum lipids cannot be determined. Second, the soil selenium content in the United States is relatively high, therefore, findings from this study may not be generalized to countries with relatively low selenium content. Finally, although confounding factors were adjusted as much as possible, there may still be some potential, unknown confounding factors that have not been considered.

## 5. Conclusions

In conclusion, the serum selenium level was positively associated with TC, TG, and LDL-C in the overall population. Meanwhile, the association between serum selenium and lipids was modified by diabetic status. Higher serum selenium levels were significantly associated with increased TC and LDL-C among participants without diabetes, but not among participants with diabetes. More studies are needed to confirm our findings and elucidate the potential mechanisms.

## Figures and Tables

**Figure 1 nutrients-15-03190-f001:**
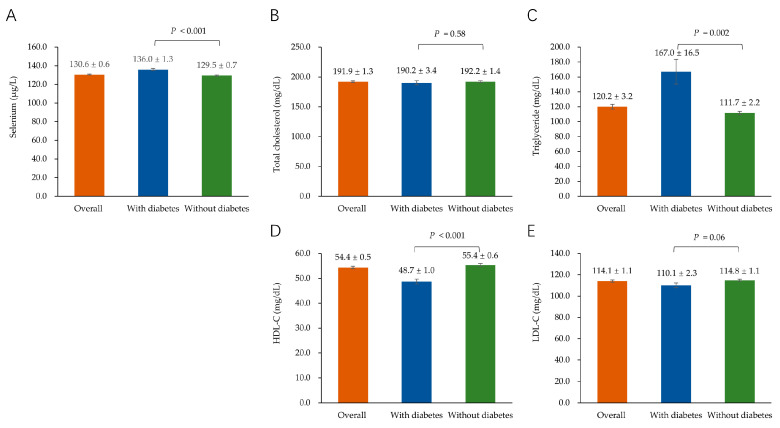
Histograms displaying means ± standard errors of serum selenium (**A**), total cholesterol (**B**), triglyceride (**C**), HDL-C (**D**), and LDL-C (**E**) levels among participants overall as well as participants with and without diabetes. *p* values refer to the comparison by one-way ANOVA between participants with and without diabetes. HDL-C, high-density lipoprotein cholesterol; LDL-C, low-density lipoprotein cholesterol.

**Table 1 nutrients-15-03190-t001:** Characteristics of the overall study population, stratified by diabetic status.

Variables	Overall Participants	Participants with Diabetes	Participants without Diabetes	*p* Value
Number	4132	782	3350	
Age, y	47.0 ± 0.5	59.1 ± 0.8	44.8 ± 0.5	<0.001
BMI, kg/m^2^	35.8 ± 2.5	44.5 ± 7.9	34.2 ± 2.5	0.13
Sex				0.10
Male	2069 (51.0)	438 (56.5)	1631 (50.0)	
Female	2063 (49.0)	344 (43.5)	1719 (50.0)	
Race/ethnicity				0.54
Mexican American	567 (8.6)	142 (8.6)	425 (8.6)	
Non-Hispanic White	1653 (66.3)	243 (66.5)	1410 (66.2)	
Non-Hispanic Black	829 (10.6)	187 (12.2)	642 (10.3)	
Others	1083 (14.6)	210 (12.8)	873 (14.9)	
Marital status				<0.001
Married	2434 (63.2)	471 (65.8)	1963 (62.7)	
Single	885 (19.8)	85 (8.8)	800 (21.8)	
Others	811 (17.0)	225 (25.4)	586 (15.5)	
Family income–poverty ratio				0.57
≤1.0	863 (15.7)	183 (17.4)	680 (15.4)	
1.0 ≤ 3.0	1537 (37.6)	305 (38.8)	1232 (37.3)	
>3.0	1383 (46.7)	204 (43.8)	1179 (47.3)	
Education level				<0.001
Less than high school	892 (16.1)	266 (20.4)	626 (15.3)	
High school or equivalent	888 (20.6)	182 (24.9)	706 (19.8)	
College or above	2350 (63.3)	332 (54.7)	2018 (64.9)	
Physical activity				<0.001
Inactive	1022 (21.0)	294 (32.5)	728 (18.9)	
Moderate	556 (13.9)	111 (16.8)	445 (13.4)	
Vigorous	2535 (65.0)	369 (50.7)	2166 (67.7)	
Smoking status				<0.001
Non-smoker	2377 (56.1)	394 (48.9)	1983 (57.5)	
Former smoker	949 (24.3)	255 (35.3)	694 (22.3)	
Current smoker	804 (19.5)	132 (15.8)	672 (20.2)	
Drinking status				0.023
Non-drinker	562 (10.9)	135 (13.0)	427 (10.5)	
Former drinker	597 (14.1)	179 (18.1)	418 (13.4)	
Current drinker	2602 (75.0)	400 (68.9)	2202 (76.2)	
HEI-2015 score	54.0 ± 0.4	54.1 ± 0.8	54.0 ± 0.4	0.91

Continuous variables are presented as means ± standard errors. Categorical variables are presented as numbers (percentages). Unweighted n, all other analyses are weighted. Differences in characteristics between subgroups were estimated using Chi-square test for categorical variables or one-way ANOVA for continuous variables. BMI, body mass index; HEI-2015, healthy eating index-2015.

**Table 2 nutrients-15-03190-t002:** Association between serum selenium and total cholesterol in the overall study population, stratified by diabetic status.

	Selenium (μg/L)					Per SD Increase	
	Tertile 1	Tertile 2		Tertile 3			
	Reference	β (95% CI)	*p* Value	β (95% CI)	*p* Value	β (95% CI)	*p* Value
Overall participants	<121.30	121.30–135.40		>135.40			
Crude	0.00	8.48 (2.60, 14.36)	0.006	14.08 (7.83, 20.33)	<0.001	6.18 (4.05, 8.31)	<0.001
Model 1	0.00	7.50 (1.72, 13.29)	0.012	13.18 (6.72, 19.65)	<0.001	5.97 (3.82, 8.12)	<0.001
Model 2	0.00	6.65 (0.33, 12.97)	0.040	12.75 (6.02, 19.47)	<0.001	5.54 (3.34, 7.74)	<0.001
Participants with diabetes	<126.00	126.00–141.20		>141.20			
Crude	0.00	−10.48 (−25.24, 4.29)	0.16	−9.97 (−27.04, 7.10)	0.25	−2.38 (−8.71, 3.94)	0.45
Model 1	0.00	−6.67 (−20.47, 7.12)	0.34	−5.85 (−22.90, 11.21)	0.49	−0.83 (−7.03, 5.37)	0.79
Model 2	0.00	−6.16 (−20.45, 8.14)	0.39	−4.41 (−22.79, 13.97)	0.63	−0.76 (−7.42, 5.90)	0.82
Participants without diabetes	<120.50	120.50–133.90		>133.90			
Crude	0.00	8.38 (2.04, 14.72)	0.011	19.27 (12.65, 25.90)	<0.001	7.69 (5.35, 10.03)	<0.001
Model 1	0.00	7.27 (1.10, 13.45)	0.022	18.03 (11.53, 24.53)	<0.001	7.33 (5.11, 9.56)	<0.001
Model 2	0.00	6.41 (−0.26, 13.08)	0.06	17.88 (10.89, 24.87)	<0.001	6.89 (4.51, 9.26)	<0.001

Data are beta coefficients (95% CIs) calculated by multiple linear regression. Results are adjusted for fasting sample weights. Model 1: adjusted for age, sex, and race/ethnicity. Model 2: model 1+ education, family income to poverty ratio, marital status, BMI, smoking status, drinking status, physical activity, and HEI-2015.

**Table 3 nutrients-15-03190-t003:** Association between serum selenium and triglyceride in the overall study population, stratified by diabetic status.

	Selenium (μg/L)					Per SD Increase	
	Tertile 1	Tertile 2		Tertile 3			
	Reference	β (95% CI)	*p* Value	β (95% CI)	*p* Value	β (95% CI)	*p* Value
Overall participants	<121.30	121.30–135.40		>135.40			
Crude	0.00	19.02 (3.15, 34.88)	0.020	29.75 (16.58, 42.92)	<0.001	12.54 (6.62, 18.46)	<0.001
Model 1	0.00	13.73 (−1.43, 28.90)	0.07	22.50 (8.68, 36.33)	0.002	9.38 (3.46, 15.30)	0.003
Model 2	0.00	9.17 (−5.15, 23.50)	0.20	23.31 (8.22, 38.40)	0.003	8.85 (2.68, 15.01)	0.006
Participants with diabetes	<126.00	126.00–141.20		>141.20			
Crude	0.00	4.68 (−64.29, 73.65)	0.89	33.09 (−4.16, 70.34)	0.08	20.63 (0.09, 41.17)	0.049
Model 1	0.00	8.12 (−61.70, 77.93)	0.82	25.47 (−10.95, 61.90)	0.17	19.08 (−1.14, 39.29)	0.06
Model 2	0.00	−21.27 (−50.93, 8.39)	0.16	38.90 (−6.77, 84.57)	0.09	21.85 (−3.41, 47.10)	0.09
Participants without diabetes	<120.50	120.50–133.90		>133.90			
Crude	0.00	10.39 (−0.32, 21.11)	0.06	26.00 (11.77, 40.22)	<0.001	8.19 (3.43, 12.96)	0.001
Model 1	0.00	5.24 (−5.45, 15.94)	0.33	20.39 (6.34, 34.44)	0.005	5.63 (0.79, 10.47)	0.024
Model 2	0.00	4.04 (−8.23, 16.32)	0.51	22.39 (6.92, 37.86)	0.006	5.57 (0.44, 10.70)	0.034

Data are beta coefficients (95% CIs) calculated by multiple linear regression. Results are adjusted for fasting sample weights. Model 1: adjusted for age, sex, and race/ethnicity. Model 2: model 1+ education, family income to poverty ratio, marital status, BMI, smoking status, drinking status, physical activity, and HEI-2015.

**Table 4 nutrients-15-03190-t004:** Association between serum selenium and high-density lipoprotein cholesterol in the overall study population, stratified by diabetic status.

	Selenium (μg/L)					Per SD Increase	
	Tertile 1	Tertile 2		Tertile 3			
	Reference	β (95% CI)	*p* Value	β (95% CI)	*p* Value	β (95% CI)	*p* Value
Overall participants	<121.30	121.30–135.40		>135.40			
Crude	0.00	−2.07 (−4.45, 0.30)	0.09	−1.31 (−3.60, 0.99)	0.26	−0.46 (−1.39, 0.47)	0.32
Model 1	0.00	−1.27 (−3.68, 1.14)	0.30	−0.07 (−2.60, 2.45)	0.95	0.19 (−0.76, 1.14)	0.69
Model 2	0.00	−1.32 (−3.83, 1.19)	0.30	−0.64 (−3.48, 2.20)	0.65	−0.03 (−1.05, 0.99)	0.95
Participants with diabetes	<126.00	126.00–141.20		>141.20			
Crude	0.00	3.66 (−1.28, 8.60)	0.14	−1.76 (−5.97, 2.44)	0.40	−0.21 (−2.10, 1.67)	0.82
Model 1	0.00	3.93 (−0.45, 8.30)	0.08	0.59 (−3.35, 4.53)	0.76	0.50 (−1.11, 2.10)	0.53
Model 2	0.00	3.02 (−1.82, 7.86)	0.22	−0.50 (−4.95, 3.95)	0.82	0.44 (−1.41, 2.28)	0.64
Participants without diabetes	<120.50	120.50–133.90		>133.90			
Crude	0.00	−2.36 (−4.67, −0.04)	0.046	−0.47 (−3.01, 2.06)	0.71	−0.15 (−1.13, 0.84)	0.76
Model 1	0.00	−1.29 (−3.47, 0.90)	0.24	0.60 (−2.01, 3.21)	0.65	0.44 (−0.52, 1.41)	0.36
Model 2	0.00	−0.95 (−3.31, 1.41)	0.42	−0.16 (−3.14, 2.82)	0.91	0.10 (−0.94, 1.14)	0.85

Data are beta coefficients (95% CIs) calculated by multiple linear regression. Results are adjusted for fasting sample weights. Model 1: adjusted for age, sex, and race/ethnicity. Model 2: model 1+ education, family income to poverty ratio, marital status, BMI, smoking status, drinking status, physical activity, and HEI-2015.

**Table 5 nutrients-15-03190-t005:** Association between serum selenium and low-density lipoprotein cholesterol in the overall study population, stratified by diabetic status.

	Selenium (μg/L)					Per SD Increase	
	Tertile 1	Tertile 2		Tertile 3			
	Reference	β (95% CI)	*p* Value	β (95% CI)	*p* Value	β (95% CI)	*p* Value
Overall participants	<121.30	121.30–135.40		>135.40			
Crude	0.00	7.66 (2.66, 12.65)	0.003	9.63 (4.73, 14.53)	<0.001	4.22 (2.13, 6.30)	<0.001
Model 1	0.00	6.81 (1.74, 11.89)	0.010	8.78 (3.50, 14.06)	0.002	3.91 (1.77, 6.06)	<0.001
Model 2	0.00	6.09 (0.79, 11.40)	0.025	8.87 (3.29, 14.46)	0.003	3.80 (1.52, 6.09)	0.002
Participants with diabetes	<126.00	126.00–141.20		>141.20			
Crude	0.00	−9.51 (−22.03, 3.01)	0.13	−14.07 (−27.42, −0.71)	0.039	−6.00 (−10.80, −1.19)	0.016
Model 1	0.00	−6.86 (−19.18, 5.46)	0.27	−11.55 (−25.68, 2.57)	0.11	−4.90 (−9.93, 0.13)	0.06
Model 2	0.00	−3.86 (−16.11, 8.38)	0.53	−9.45 (−23.11, 4.21)	0.17	−4.51 (−9.27, 0.25)	0.06
Participants without diabetes	<120.50	120.50–133.90		>133.90			
Crude	0.00	8.29 (3.16, 13.42)	0.002	14.61 (9.51, 19.71)	<0.001	6.11 (3.83, 8.39)	<0.001
Model 1	0.00	7.08 (1.92, 12.24)	0.008	13.29 (8.03, 18.55)	<0.001	5.64 (3.39, 7.89)	<0.001
Model 2	0.00	5.94 (0.33, 11.55)	0.038	13.43 (7.68, 19.18)	<0.001	5.51 (3.02, 8.00)	<0.001

Data are beta coefficients (95% CIs) calculated by multiple linear regression. Results are adjusted for fasting sample weights. Model 1: adjusted for age, sex, and race/ethnicity. Model 2: model 1+ education, family income to poverty ratio, marital status, BMI, smoking status, drinking status, physical activity, and HEI-2015.

**Table 6 nutrients-15-03190-t006:** The interaction between serum selenium and diabetic status for serum lipids.

	TC	TG	HDL-C	LDL-C
Crude	0.002	0.30	0.97	<0.001
Model 1	0.003	0.30	0.80	<0.001
Model 2	0.007	0.39	0.64	<0.001

The interaction indicated per standard deviation (SD) increase in serum selenium × diabetic status. The serum selenium was treated as a continuous variable in the interaction analysis. Results are adjusted for fasting sample weights. Model 1: adjusted for age, sex, and race/ethnicity. Model 2: model 1+ education, family income to poverty ratio, marital status, BMI, smoking status, drinking status, physical activity, and HEI-2015.

## Data Availability

Publicly available datasets were analyzed in this study. These data are available from the National Health and Nutrition Examination Survey.

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
