# Peer review of "Association between Serum Selenium Levels and Lipids among People with and without Diabetes"

_nutrients, 2023, doi:10.3390/nu15143190_

Round 1

Reviewer 1 Report

Although the study titled ’Association between serum selenium levels and lipids among people with and without diabetes is interesting, I still have some comments.

Introduction is complex, but some information which is in discussion section must be implemented into introduction, such as the association between selenium and lipids, the mechanisms – lines 257-274, should be a bit shortened. This is not suitable for discussion section.

The scientific language should be improved, e.g. lines 29-32 essential trace elements are repeated 3 times; line 25 people living with – this is unacceptable, line 43 subjects; the word ’abnormal’ should be changed into e.g. impaired, the word ’abnormal’ is used too often; diabetes status should be changes into diabetic status; lines 242-243 consistent, inconsistent – one of these words must be changed.

Definition of diabetes cannot be acceptable in this form. There are recommendations to diagnose diabetes and authors should based on them e.g. Diabetes Care 2019;42(Suppl. 1):S13–S28 | https://doi.org/10.2337/dc19-S002; we cannot diagnose diabetes basing on ’told that they had diabetes or are taking any antidiabetic medication or insulin’. Also the proper reference must be included.

Line 111 – Healthy eating index-2015 (HEI-2015), please add reference, was it validated? Readers not necessarily should know what it is.

Line 105: physical activity is not well defined; it should be more specific, e.g. how many times a week for how long, in this form it is not acceptable; what is it moderate recreational activities or vigorous?

The results are well presented in the text, but tables 2,3,4,5 are not clear for a reader; please modify them.

Do not add the strenghts of your manuscript, please, delay 284-291 lines.

-

Author Response

请参阅附件。

Reviewer 2 Report

The manuscript prepared by Nie Qi et al.  presented the results of the study focused on exploration of the association between serum selenium and lipids among subjects with diabetes and without diabetes.

Major comments:

11. In the introduction section, there is no information about diabetes and the relationship between diabetes and CVD.

22.  In the study design and participants section, please indicate precisely the inclusion and exclusion criteria for the study.

33. How long were the serum samples stored at – 20° C before assaying. Do you think that serum samples should be stored at -80° C to maintain their stability?

44.  The results section should start with the presentation of the serum selenium and serum lipids levels in the individual groups on the graphs. Examples: Figure 1 – serum selenium level; Figure 2 – serum lipids level.

55. In the conclusion section, it should be clarified how the relationship between the serum between serum selenium and lipids was modified by diabetes status based on the results of presented in the study.

Round 2

Reviewer 1 Report

Authors answered all my questions and the manuscript was improved.

Minor English improvement required.

Reviewer 2 Report

Accept in present form.